# Real-Life Prognosis of Sinonasal Tumors

**DOI:** 10.3390/jpm14050444

**Published:** 2024-04-24

**Authors:** Giancarlo Pecorari, Gian Marco Motatto, Federica Piazza, Alessandro Garzaro, Giuseppe Riva

**Affiliations:** Division of Otorhinolaryngology, Department of Surgical Sciences, University of Turin, 10126 Turin, Italy; giancarlo.pecorari@unito.it (G.P.); gmotatto@cittadellasalute.to.it (G.M.M.); f.piazza@unito.it (F.P.); alessandro.garzar@edu.unito.it (A.G.)

**Keywords:** sinonasal cancer, survival, nasal tumors, real-life prognosis

## Abstract

Background: Sinonasal cancer represents a challenging disease because of its difficult diagnosis and different histology. Despite a multidisciplinary evaluation and treatments, a poor prognosis is still present. We retrospectively analyzed patients with sinonasal cancer treated in our institution, paying attention to histology and real-life prognosis. Methods: A total of 51 consecutive patients were included in the study. Clinical features were described. Overall, disease-free, and disease-specific survival (OS, DFS, DSS) according to histology were calculated. Kaplan–Meyer estimator curves were reported. Results: The most prevalent primary tumor was squamous cell carcinoma, followed by adenocarcinoma. Global 2- and 5-year OS was 68.80% and 54.58%, respectively. Global 2- and 5-year DFS was 48.53% and 29.56%, while global 2- and 5-year DSS was 82.86% and 74.57%, respectively. The median OS was 74 and 43 months for early- and late-stage cancer, respectively. The Cox multivariate regression analysis did not reveal any statistically significant effects of age, stage, or histology on survival outcomes. Conclusions: The diagnosis is often late and the prognosis poor. An appropriate treatment, which is always quite multimodal, allows us to achieve a global 5-year OS slightly higher than 50%. An adequate diagnosis to increase the percentage of early-stage tumors is mandatory to improve prognosis.

## 1. Introduction

Sinonasal malignancies are a group of rare tumors characterized by histopathological heterogeneity [1]. They include some histotypes which are exclusive of the sinonasal tract, such as sinonasal adenocarcinoma, sinonasal undifferentiated carcinoma, sinonasal squamous cell carcinoma, and olfactory neuroblastoma. In addition, other malignancies that are seen in the sinonasal region are included, such as mucosal melanoma, nuclear protein of the testis (NUT) carcinoma, and extranodal natural killer cell (NK)/T-cell lymphoma [2]. Globally, the most common entities are squamous cell carcinoma (SCC), adenocarcinoma and mucosal melanoma [1]. Given the rarity and the heterogeneity of histology, real-life studies of prognosis are needed.

Heterogeneous biological behavior frequently presents a challenge in differential diagnosis and treatment choice [3,4]. The prognosis depends on the initial tumor stage and histological subtype [5,6]. It is important to underline that despite the evolvement of new treatment strategies in the past decade, specifically advanced transnasal endoscopic surgical techniques and high-precision radiotherapy, the outcomes of sinonasal tumors have remained relatively stable [7]. Real-life data are needed to improve daily clinical practice. Indeed, in contrast to clinical trials, physicians had to adequately treat patients with comorbidities and complex health problems.

Poor outcomes are also related to a late diagnosis. Indeed, the diagnosis is often made when the tumor is at a local advanced or metastatic stage [8,9]. More than 80% of sinonasal SCCs are diagnosed at stage T3 or T4 due to non-specific symptoms [10]. The delay is due to non-specific early symptoms which are common in benign diseases, like facial pain, chronic rhinorrhea, epistaxis, and nasal obstruction [11,12]. Proptosis, diplopia, cranial neuropathy, or headaches are associated with a locally advanced disease. On the other hand, the clinical manifestation of an advanced disease is related to the location and extension of the neoplasia. Anosmia and proptosis are associated with an extension toward the anterior cranial fossa through the cribriform plate or the orbit. Secondly, if the middle cranial fossa is invaded, patients report paresthesia of the lower face or trismus due to the involvement of cranial nerve V3 or an invasion of the pterygoid muscles. Lastly, extension to the lateral bones and invasion of the cavernous sinuses determine the neuropathy of cranial nerves III, IV, VI, V1, and V2, leading to diplopia and paresthesia of the face [8].

The aim of this retrospective observational study was to evaluate prognosis of malignant sinonasal cancer in a real-life setting. In particular, a comparison of survival rates of different tumors was performed.

## 2. Materials and Methods

From January 2015 to June 2023, 51 patients with malignant sinonasal tumor were diagnosed and treated in our department and included in the study. In particular, inclusion criteria regarded all the consecutive subjects diagnosed and treated in our department in order to perform a real-life study without excluding some patients because of comorbidities, such as in clinical trials. Exclusion criteria were benign tumors and sinonasal metastases. A chart review was performed to collect clinical data (age, sex, histology, tumor stage and grading, treatment, recurrence, and death). The study was conducted in accordance with the Declaration of Helsinki and approved by the Institutional Review Board (protocol code 0090416, date of approval 28 July 2023). Written informed consent was obtained.

World Health Organization classification of sinonasal tumors (5th edition, 2022) was used [13]. The American Joint Committee of Cancer (AJCC) staging system, 8th edition (TNM—Tumor Node Metastasis), was used for epithelial tumors, the Ann Arbor classification for lymphomas, the Revised multiple Myeloma International Staging System (R-ISS) for plasmacytoma in multiple myeloma, and the Kadish for olfactory neuroblastoma. All the patients underwent an endoscopic biopsy under local or general anaesthesia. The treatment was based on national and international guidelines.

All statistical analyses were carried out using a Statistical Package for Social Sciences, version 20.0 (IBM Corporation, Armonk, NY, USA), and GraphPad Prism, version 9 (GraphPad Software Inc., San Diego, CA, USA). A descriptive analysis of all data was performed, and they were reported as means, medians or percentages and standard deviations. A Kolmogorov–Smirnov test demonstrated a non-Gaussian distribution of variables, so non-parametric tests were used. A Friedman F test Mann-Whitney U test was used to assess differences between groups in the mean of continuous variables, while a chi-squared test was used for categorical variables. The Kaplan-Meier method was used for overall survival (OS), disease-free survival (DFS) and disease-specific survival (DSS), and a log-rank test was used for univariate survival analyses. The endpoints were the length of time from diagnosis to death by any cause for OS, from diagnosis to recurrence or death for DFS, and from diagnosis to death for cancer for DSS. Cox proportional hazards regressions were used for univariate and multivariate survival analyses to concurrently control for variables such as age, tumor stage, and pathology type. Only groups with at least five subjects with the same histology were included in the Cox regression. A *p* value less than 0.05 was considered statistically significant.

## 3. Results

A total of 51 consecutive patients with primary tumor of the nasal cavities and/or paranasal sinuses were included in the study. The most prevalent primary tumor was squamous cell carcinoma (Figure 1). One undifferentiated carcinoma was HPV-positive.

The mean age was 68.59 ± 15.37 years (range 21–89 years). A total of 34 patients were male (66.7%). A total of 11 patients (21.5%) were smokers, while occupational exposure to wood dust was reported in eight cases (intestinal-type adenocarcinomas). TNM stage for epithelial tumors was as follows (n, %): T1 9 (21.4%), T2 7 (16.7%), T3 9 (21.4%), T4 17 (40.5%), N0 32 (76.2%), N1 2 (4.8%), N2 8 (19.0%), N3 0 (0%), M0 40 (95.2%), M1 2 (4.8%) (Table 1).

Mucous melanomas were T3 in one case and T4a in three cases and N stage was N0 in three patients and N1 in one subject. The three B cell lymphomas (two diffuse large B cell lymphomas and one small lymphocytic lymphoma) were diagnosed as stage IIE, IIIE, and IVE DLBCL according to the Ann Arbor classification, respectively. The plasmacytoma in multiple myeloma had a R-ISS stage 2. Olfactory neuroblastoma was diagnosed as a Kadish C stage.

Non-surgical treatment was adopted in 11 cases (21.6%), while 40 patients (78.4%) underwent uni- or multimodal treatment including surgery. In particular, an endoscopic approach was used in 21 cases, an open approach in 18 cases and a combined on one patient. Exenteratio orbitae was necessary in two cases. Neck dissection was performed on six patients. Concerning epithelial tumors, adjuvant radiotherapy was performed on nine patients and adjuvant chemoradiotherapy on four patients, whereas neoadjuvant chemotherapy was administered in two cases before surgery and to another two subjects before chemoradiotherapy.

At the end of the study (June 2023), 32 patients (62.7%) were alive. In particular, 22 of them (43.1%) were alive without evidence of disease, while 10 patients (19.6%) experienced a recurrence, although they were still living. Of the 19 patients who had died (37.3%), 9 (17.6%) had a recurrence of the disease while 10 (19.6%) died of other causes.

Global 2- and 5-year OS was 68.80% and 54.58%, respectively. Global 2- and 5-year DFS was 48.53% and 29.56%, while global 2- and 5-year DSS was 82.86% and 74.57%, respectively (Figure 2). Table 2 and Figure 3 highlight OS, DFS and DSS for each histology.

Concerning epithelial tumors, a non-statistically significant trend in favor of early-stage cancer was observed (*p* values for OS, DFS and DSS at the log-rank test were 0.077, 0.252, 0.465, respectively). In particular, the median OS was 74 and 43 months for early- and advanced-stage cancer, respectively. The median DFS and DSS were not estimable (more than 50% of patients were free of the disease after 5 years of follow-up), 74 months for early-stage tumors, and 18 and 84 months for advanced-stage ones (Figure 4).

The comparison between squamous cell carcinomas and adenocarcinomas did not show statistically significant differences (*p* values for OS, DFS and DSS at the log-rank test were 0.406, 0.224, and 0.108, respectively). Patients with adenocarcinoma seemed to have a better prognosis. However, we should take into consideration the higher percentage of advanced stage in SCCs compared to adenocarcinomas (36.4% and 73.7%, respectively; *p* = 0.006).

Therefore, we performed a univariate and multivariate Cox proportional hazards regressions (Table 3). No statistically significant difference was observed considering age (younger versus older than 70 years), stage (early versus late) and histology (ITAC, keratizing SCC and non-keratizing SCC) (*p* > 0.05).

## 4. Discussion

Malignant sinonasal tumors represent approximately 0.5% of all malignant neoplasms and are characterized by considerable histopathological heterogeneity [14]. They represent less than 5% of all head and neck neoplasms, with an incidence of 0.56 per 100.000 individuals per year [2]. Real-life data are crucial for improving daily clinical practice.

SCC represents approximately 3% of head and neck cancers and up to 61% of sinonasal tract, followed by adenocarcinoma and mucosal melanoma, and is more common in men and in patients over 80 years of age [15]. Tobacco and occupational exposure to wood, leather dust, glue, formal aldehyde, arsenic, chrome, nickel, and welding fumes are reported as a risk factor for sinonasal SCC [9,16,17]. Furthermore, recent studies showed an association between human papillomavirus (HPV) and sinonasal SCC, suggesting its potential causative role. The patients with HPV-positive sinonasal SCC are younger than subjects with HPV-negative SCC and have a higher 5-year overall survival rate [18,19,20]. The most frequent site of SCC is the maxillary sinus (around 60%), followed by the nasal cavity (25%) and the ethmoidal cells (15%) [21,22].

Surgery is a potentially curative treatment for SCC if a complete resection with negative margins can be performed. The surgical approach, whether endoscopic or open surgery, depends on the location and local extension of the tumor [23]. For this reason, patients should be aware that surgery could be not curative in advanced-stage due to the impossibility of achieving a total resection [2]. Adjuvant intensity-modulated radiation therapy, after a complete resection, is the standard treatment for decreasing the risk of local recurrence for pT2 through pT4 sinonasal SCCs. Platinum-based adjuvant therapy is frequently indicated with the aim of radiosensitization and targeting of residual disease in cases of positive margins with perineural, nodal or lymphovascular invasion [21]. In an unresectable tumor or in patients who cannot undergo or do not choose surgery, chemoradiation is the standard protocol [2]. For locoregional recurrence, surgery is recommended [24]. If it is not possible, palliative systemic chemotherapy is the treatment of choice [25].

The 5-year OS for SCC has been around 50% in recent decades, basically due to local recurrence [26,27,28]. Nasal cavity SCCs have an increased 5-year survival rate compared with patients with SCCs in the paranasal sinuses [5,29]. Negative prognostic factors are male sex, local advanced or metastatic disease, and older age [28]. Recurrence is as high as 50% at 5 years [30]. According to head and neck cancer guidelines, patients undergo clinical and radiologic examinations for at least 5 years due to the high recurrence risk [31].

Adenocarcinoma is the second most common histotype of sinonasal cancer, including about 27% of all sinonasal tumors [32]. More than 50% of adenocarcinomas have their origin in the ethmoid sinuses and the nasal cavity [2]. Sinonasal adenocarcinoma can be divided into salivary and non-salivary types. The latter can be further divided into intestinal and non-intestinal types [33]. Environmental exposure has not any reported association with non-intestinal adenocarcinoma. On the other hand, the intestinal-type adenocarcinoma (ITAC) has been reported to be related to long-term exposure to wood dust [33]. Moreover, in the literature there are reported cases of sinonasal renal cell-like adenocarcinoma [34].

As for SCC, a complete removal and the prevention of locoregional recurrence are the aims of the treatment in adenocarcinoma. Surgery is the first choice, with endoscopic resection having a survival advantage. However, an endoscopic approach can be chosen only for smaller and lower-stage tumors [35]. There is no recommendation for prophylactic neck dissection in adenocarcinoma. Some studies reported the use of adjuvant and neoadjuvant radiotherapy, but there are no prospective studies [2]. Several chemotherapeutic agents are reported to be effective. In particular, Roux et al. reported a 37% overall response rate in neoadjuvant protocol with cisplatin and 5-fluorouracil, while a complete response rate was observed only in 15% of cases [36]. Poor prognostic factors are paranasal sinus involvement, black race, a local advanced stage, age >75 years, and high grade [37,38]. The 5-year survival has improved in recent decades with a 5-year DSS higher than SCC [37].

Sinonasal undifferentiated carcinoma (SNUC) is a rare carcinoma, which represents only 5% of sinonasal malignancies [15]. Originally, it was described as an aggressive carcinoma that originated from the Schneiderian epithelium and/or nasal ectoderm [2]. SNUCs have a poor prognosis; they tend to be recurrent and patients often develop metastases, leading to death. The average survival is 22 months with a 5-year survival rate of 34.9% [39].

Primary mucosal melanoma is rare and only 1.3% of all melanomas have a mucosal origin. Approximately 70% of mucosal melanomas are located in the head and neck district and the most common head and neck locations are the nasal cavity and paranasal sinuses [40,41]. The incidence is lower than in cutaneous or ocular melanoma [41]. It is more common in Caucasians and older patients, with no sex difference, while the average age at diagnosis is 70 years [41].

The average OS for mucosal melanoma is 26 months and the 5-year survival rate is about 22% [42]. The prognosis is worse than for cutaneous melanoma, and a paranasal localization is related to lower survival due to the fact that paranasal melanomas are more advanced when diagnosticated than nasal cavity ones [41,43]. Because of a local advanced disease (i.e., infiltration of the skull base, orbit or facial tissue), ethmoidal and maxillary sinus melanomas have the worst prognosis [44]. Early and repeated recurrences are frequent [41].

In our study, a cohort of 51 consecutive patients affected by de novo sinonasal malignancies was diagnosed and treated at a single tertiary academic referral center over a period of 8 years. Our sample was quite wide when compared with other studies with a similar duration, confirming that sinonasal malignant neoplasms are rare entities [45,46].

As expected from the literature, there was a wide heterogeneity in histopathological findings with SCC as the most represented histotype, followed by adenocarcinoma and mucosal melanoma [2,32]. In our study, SCC represented 37.25% of cases, while adenocarcinoma 21.57% of subjects. In a study by Robin et al., based on a pool of 11,160 patients from the National Cancer Database, the prevalence of SCC and adenocarcinoma was 54% and 7%, respectively [5]. The differences between our study and the literature can be explained both by greater occupational exposure to wood and leather dust (an important risk factor for adenocarcinoma) and by the reduced sample size [1].

SCC and adenocarcinoma constitute approximately 65–75% of all sinonasal malignant tumors, while the sum of all other histological variants constitutes 25–35% of the total [32]. Given these data, the difficulty in collecting information both from the population and from the literature is noticeable, in particular for the less frequent or more recent histotypes introduced by the WHO classification [13].

The stage at diagnosis has an early-to-late ratio of 1:2 [1]. The main reason is the difficulty in performing a correct diagnosis at an early stage because of absent or non-specific symptoms that are common in benign nasal complaints. Furthermore, the use of antibiotic therapies may lead to a temporary regression of symptoms, falsely reassuring both the physician and the patient [11,12].

In our series, the global 2-year OS was 68.8%, with a wide range of variation depending on the histotype. The OS of this study seems to be longer as compared to other studies, especially when compared with older ones, and quite similar to the new series [47,48,49,50]. We found a global 2-year DSS of 82.9%. Concerning the histotype, adenocarcinoma had a better 2-year DSS (100%) than SCC. Despite a non-statistically significant difference, patients with adenocarcinoma had a better prognosis. The reason for this is the higher percentage of early stages in adenocarcinomas. In comparison, the 5-year DSS of 74.6% for patients in our cohort was higher compared to the two registry-based studies reported (57–63%) [47,49].

The same applies to the DFS. In our cohort of patient, global 2-year DFS was 48.5%, with a non-statistically significant difference between adenocarcinoma and squamous cell carcinoma.

In 2022, Hafstrom et al. performed a population-based cohort analysis of 226 consecutive patients from Northern Europe affected by sinonasal malignances. A total of 120 patients (53.1%) suffered from squamous cell carcinoma, 29 (12.8%) from mucosal melanoma, 23 (10.2%) from adenocarcinoma, 15 (6.6%) from olfactory neuroblastoma, 13 (5.8%) from undifferentiated carcinoma, 5 (2.2%) from neuroendocrine carcinoma, and 1 (0.4%) from NUT carcinoma. The 5-year OS for all patients was 57%, quite similar to our study. Five-year OS, DFS and DSS for patients treated with curative intent were, respectively, 70.2%, 63.4%, and 82.8%. In these cases, the authors were able to demonstrate the impact of stage, histopathology, treatment modality and primary site on DSS, identifying patients affected by adenocarcinoma and salivary carcinoma as those with higher DSS [50]. In our study, we compared the survival rates of early- and late-stage patients, but we were not able to demonstrate a statistically significant difference (*p* values for OS, DFS and DSS at the log-rank test were 0.406, 0.224, and 0.108, respectively), even if patients with early-stage neoplasms seemed to have a longer survival.

Due to the relatively low number of cases of other histotypes, we focused on survival of SCC and adenocarcinoma. Two-year OS, DFS, and DSS for the former and the latter were, respectively, 76.6%, 45.8%, and 87.2%, and 90.9%, 68.6%, and 100%. However, even though graphs showed a more favorable trend for adenocarcinoma, no statistically significant differences were found between the two groups. This is due to the higher percentage of patients with SCC diagnosed at an advanced stage compared to adenocarcinoma.

In a North American study by Jain et al. on 2.895 SCCs and 819 adenocarcinomas, the 5-year OS was significantly higher in adenocarcinoma (67.4%) compared to SCC (58.6%). Five-year DSS was 39.1% for SCC and 57.5% for adenocarcinoma [27]. These results seemed to be worse than our study. A possible explanation could be the long period of data collection (from 1973 to 2012) in the study by Jain et al. and due to a smaller sample in our study.

The strength of this study was the inclusion of consecutive patients referring to our tertiary center with a new diagnosis of sinonasal cancer, in order to avoid selection bias. Since all consecutive patients with malignant sinonasal tumor were included, the risk of selection bias was minimal. This allowed us to obtain real-life data, without excluding subjects with particular comorbidities or tumor histology, in contrast to clinical trials. Another strength was the inclusion of patients who underwent both a surgical and a chemoradiation treatment. Moreover, both open and endoscopic approaches were considered.

The main limit of our study is the relatively small sample, if we consider the great heterogeneity. Indeed, some histotypes are represented by only a few patients. Future studies with real-life data are mandatory to perform more robust multivariate analyses.

However, it is important for clinical practice to stress the importance of an ade-quate and early diagnosis of sinonasal tumors to improve the outcome.

The clinical examination of patients with suspected sinonasal cancer should start with a medical history and a complete ear, nose, and throat exploration, including evaluation of the neck and the cranial nerves. Moreover, nasal endoscopy is mandatory due to the limited information provided by a simple anterior rhinoscopy. Imaging tests are essential in order to achieve a correct diagnosis since they allow us to evaluate the complete extent of the tumor and can potentially provide information on its benign or malignant nature. Today, when strong clinical evidence of malignancy is present, both CT and MRI with contrast enhancement are performed in order to obtain more details about the localization and local extension of the neoplasm, and neck or distant-site involvement. Indeed, both are essential to determine operability and to plan radiotherapy [37].

Based on our data and the literature, postoperative surveillance is essential for these patients. MRI is considered the standard imaging for a follow-up. According to the literature, after treatment of the primary tumor, 10% of patients with sinonasal tumors develop distant metastasis; however, this rarely occurs in the absence of locoregional recurrence [37].

## 5. Conclusions

Sinonasal cancer includes several tumors with a low incidence and non-specific symptoms. The diagnosis is often late and the prognosis poor. An appropriate treatment, which is quite always multimodal, allows us to achieve a global 2-year OS of about 70%. Recurrences are common, determining a global 2-year DFS of about 50%. On the other hand, the global 2-year DSS is about 80%, suggesting that these patients do not die as often because of cancer. An adequate diagnosis to increase the percentage of early-stage tumors is mandatory to improve prognosis.

## Figures and Tables

**Figure 1 jpm-14-00444-f001:**
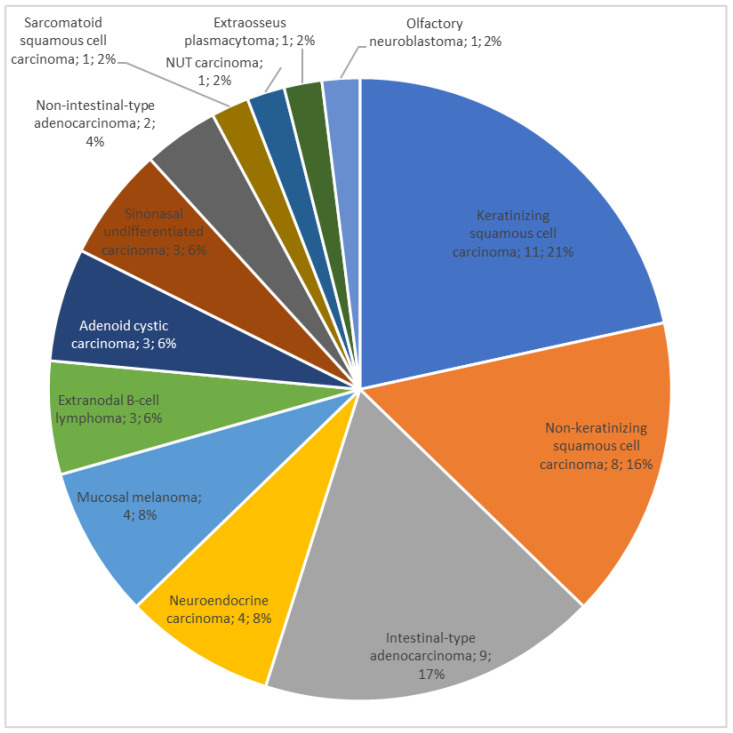
Primary tumors (n, %) of the sinonasal region.

**Figure 2 jpm-14-00444-f002:**
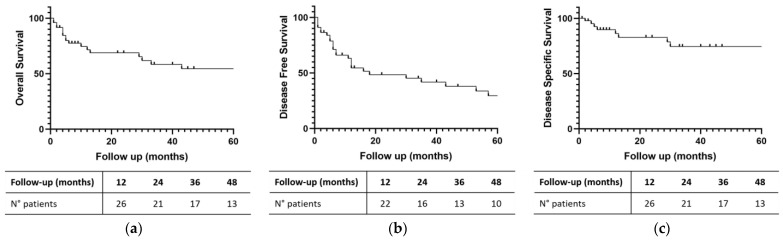
(**a**) Global Overall Survival; (**b**) Global Disease-Free Survival; (**c**) Global Disease-Specific Survival for sinonasal tumors. Number of patients at follow-up intervals are displayed.

**Figure 3 jpm-14-00444-f003:**
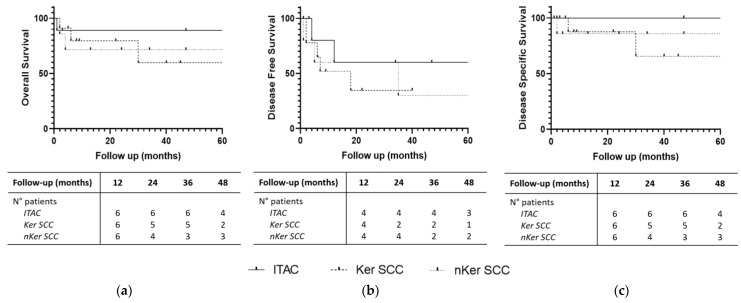
(**a**) Overall Survival; (**b**) Disease-Free Survival; (**c**) Disease-Specific Survival for different histology. Groups with fewer than five subjects are not displayed. Number of patients at follow-up intervals are displayed.

**Figure 4 jpm-14-00444-f004:**
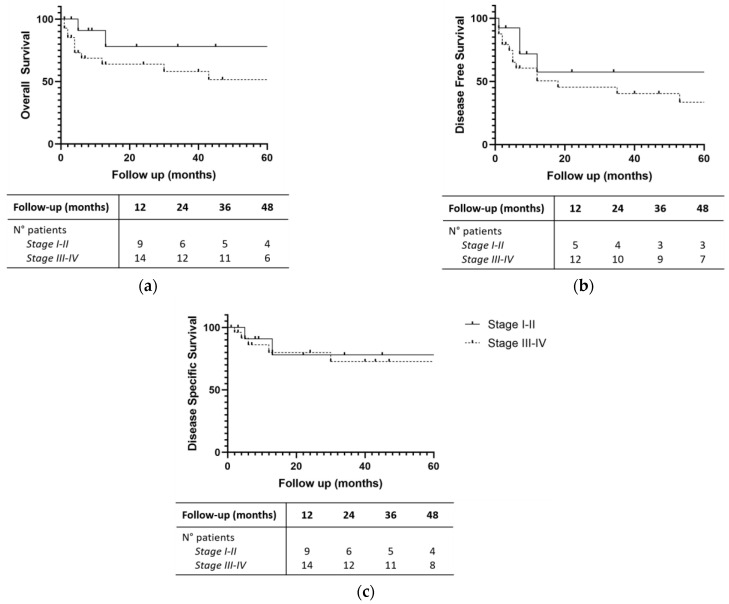
(**a**) Overall Survival; (**b**) Disease-Free Survival; (**c**) Disease-Specific Survival for early- and advanced-stage epithelial tumors. Number of patients at follow-up intervals are displayed.

**Table 1 jpm-14-00444-t001:** Clinical characteristics.

Characteristics	
Age (years)	
Mean ± Standard deviation	68.59 ± 15.37
Range	21–89
Sex (n; %)	
Male	34; 66.7%
Female	17; 33.3%
Smoker (n; %)	
Yes	11; 21.5%
No	40; 78.5%
Exposure to wood dust (n; %)	
Yes	8; 15.7%
No	43; 84.3%
Stage (n; %)	
I	7; 16.7%
II	7; 16.7%
III	9; 21.4%
IV	19; 45.2%

**Table 2 jpm-14-00444-t002:** Median survival, 2- and 5-year OS, DFS and DSS for each histology.

Histology (n°)	Median Follow-Up (Months)	Median Survival (Months)	2-Year OS	5-Year OS	2-Year DFS	5-Year DFS	2-Year DSS	5-Year DSS
Keratinizing SCC (11)	8	84	79.54	59.65	34.56	-	87.50	65.62
Non-keratinizing SCC (8)	24	>60	71.42	71.42	60.00	30.00	85.71	85.71
Sarcomatoid SCC (1)	12	12	0.00	0.00	0.00	0.00	0.00	0.00
ITAC (9)	3	74	88.88	88.88	60.00	60.00	100.00	100.00
Non-ITAC (2)	35	>60	100.00	100.00	100.00	100.00	100.00	100.00
SNUC (3)	74	>60	100.00	100.00	100.00	100.00	100.00	100.00
NUT carcinoma (1)	6	- *	100.00	-	0.00	-	100.00	-
NEC (4)	6	9	0.00	0.00	0.00	0.00	0.00	0.00
ACC (3)	4	4	33.33	0.00	33.33	0.00	66.67	0.00
Mucosal melanoma (4)	17	29	66.67	33.33	25.00	0.00	100.00	50.00
B cell lymphoma (3)	10	35	50.00	50.00	50.00	0.00	100.00	50.00
Plasmacytoma (1)	13	- *	100.00	-	100.00	-	100.00	-
Olfactory neuroblastoma (1)	33	33	100.00	0.00	100.00	0.00	100.00	0.00

ACC, adenoid cystic carcinoma; ITAC, intestinal-type adenocarcinoma; NEC, neuroendocrine carcinoma; SCC, squamous cell carcinoma, SNUC, sinonasal undifferentiated carcinoma. * Undefined due to censored patients.

**Table 3 jpm-14-00444-t003:** Univariate and multivariate Cox proportional hazard regressions.

Variables	Univariate		Multivariate	
	HR (95% CI)	*p* Value	HR (95% CI)	*p* Value
** *Overall Survival* **				
Age <70 years >70 years	1 (reference) 0.431 (0.079–2.336)	0.303	1 (reference) 0.319 (0.038–2.072)	0.235
Stage Early (I–II) Late (III–IV)	1 (reference) <0.001 (N/A)	0.999	1 (reference) <0.001 (N/A)	0.999
Histology ITAC Keratinizing SCC Non-keratinizing SCC	1 (reference) 1.872 (0.343–14.21) 1.316 (0.153–11.29)	0.486 0.787	1 (reference) 0.593 (0.066–5.157) 0.296 (0.023–3.131)	0.612 0.304
** *Disease Free Survival* **				
Age <70 years >70 years	1 (reference) 0.306 (0.084–1.110)	0.063	1 (reference) 0.367 (0.075–1.554)	0.179
Stage Early (I–II) Late (III–IV)	1 (reference) 2.771 (0.702–18.32)	0.195	1 (reference) 2.343 (0.554–15.95)	0.294
Histology ITAC Keratinizing SCC Non-keratinizing SCC	1 (reference) 2.010 (0.463–10.66) 1.462 (0.269–7.947)	0.365 0.643	1 (reference) 1.173 (0.192–7.216) 0.811 (0.119–5.027)	0.857 0.819

CI, confidence interval; HR, hazard ratio; ITAC, intestinal-type adenocarcinoma; N/A, not applicable; SCC, squamous cell carcinoma.

## Data Availability

The data presented in this study are available on request from the corresponding author.

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
