# Peer review of "Real-Life Prognosis of Sinonasal Tumors"

_jpm, 2024, doi:10.3390/jpm14050444_

Round 1

Reviewer 1 Report

Comments and Suggestions for Authors

Major consideration:

This retrospective observational study endeavors to retrospectively assess the prognostic outcomes of malignant sinonasal cancer within a single institution. Nonetheless, the manuscript does not adequately delineate the existing knowledge gap or hypothesize effectively, resulting in an ambiguous analytical trajectory. Presuming the objective is to explore the determinants influencing survival rates in sinonasal carcinoma, it becomes imperative to concurrently control for or examine variables such as age, tumor stages, and pathology types. This approach necessitates the application of multivariable survival analysis to ensure a comprehensive and accurate evaluation of prognostic factors.

Abstract

The conclusion should be more succinct, focusing on the principal findings and their implications.

Introduction

1.     Emphasize the significance of the research topic in the initial paragraph to highlight its importance.

2.     Clearly articulate the knowledge gap that motivated this study, aligning it with the study's objectives.

Methods

1.     Recommend employing Cox regression for both univariate and multivariate survival analyses if the assumptions are met. Relying solely on the Kaplan-Meier method and the Log-rank test may be insufficient.

2.     The justification for not performing a multivariate analysis due to the small sample size is unconvincing. Important confounders affecting survival, such as age and stage, should be considered.

Results

1.     In Figure 1, include the number of patients of subtype.

2.     Clarify the data presented in Line 97. (specifically T1 9 (19.6%).?)

3.     A table summarizing clinical characteristics is essential.

4.     Present data on the number of recurrences, deaths, and cause-specific deaths.

5.     In Figure 2, include the number of patients at specific follow-up intervals (e.g., 12, 24, 36 months).

6.     Concerns regarding Table 1:

I.         Clarify the statement regarding ">60 months median survival." Given the study commenced in January 2015, and the longest follow-up was approximately 9 years (108 months), a median survival time of 160 months seems inaccurate.

II.        Consider presenting the median follow-up time and outcome numbers for each subgroup instead.

7.     For Figure 3:

I.         Avoid displaying results for subtypes with fewer than five patients to prevent confusion.

II.        Include patient counts at specific follow-up intervals for each group.

8.     Lines 137-139 suggest a misunderstanding of median OS, median DSS, and median DFS. Please review these definitions.

9.     For Figure 4, indicate patient follow-up numbers for each group at specific time points.

10.  Regarding Figure 5:

I.         Correct the legend for Figure 5.

II.        If Figure 5 is largely redundant with Figure 3, consider simplifying or omitting it.

11.  Lines 154-156: Given these acknowledgments, multivariable Cox regression analysis becomes necessary for meaningful comparisons.

Discussion

1.     Summarize the key findings with reference to the study's objectives.

2.     The discussion should be focused, driven by a central thesis based on the study's primary objectives and findings.

3.     Consider including the following reference for discussion:

“Wu CN, Chuang IC, Chuang MJ, Chen WC. "Sinonasal renal cell-like adenocarcinoma: An Easily Misdiagnosed Sinonasal Tumor." This could provide valuable context and comparison to your findings.

Author Response

Dear Reviewer,

We thank you for the opportunity to revise our paper. Hereby you will find attached a point-by-point answer to each suggestion. Changes were highlighted.

  • Major consideration: This retrospective observational study endeavors to retrospectively assess the prognostic outcomes of malignant sinonasal cancer within a single institution. Nonetheless, the manuscript does not adequately delineate the existing knowledge gap or hypothesize effectively, resulting in an ambiguous analytical trajectory. Presuming the objective is to explore the determinants influencing survival rates in sinonasal carcinoma, it becomes imperative to concurrently control for or examine variables such as age, tumor stages, and pathology types. This approach necessitates the application of multivariable survival analysis to ensure a comprehensive and accurate evaluation of prognostic factors.
  • Thanks for your comments and suggestions. A multivariate survival analysis (Cox proportional hazards regressions) was performed and described.

  • Abstract: The conclusion should be more succinct, focusing on the principal findings and their implications.
  • The conclusion was shortened and focused on the main findings.

  • Introduction:
  1. Emphasize the significance of the research topic in the initial paragraph to highlight its importance.
  • The significance of the research topic was highlighted in the initial paragraph.

  • Clearly articulate the knowledge gap that motivated this study, aligning it with the study's objectives.
  • We better described the knowledge gap that motivated the study.

  • Methods:
  1. Recommend employing Cox regression for both univariate and multivariate survival analyses if the assumptions are met. Relying solely on the Kaplan-Meier method and the Log-rank test may be insufficient.
  2. The justification for not performing a multivariate analysis due to the small sample size is unconvincing. Important confounders affecting survival, such as age and stage, should be considered.
  • The Cox regression was performed for multivariate survival analyses and described. Age, stage and histology were considered.

  • Results:
  1. In Figure 1, include the number of patients of subtype.
  • The number of patients were added to figure 1.

  • Clarify the data presented in Line 97. (specifically T1 9 (19.6%).?)
  • Data were clarified.

  • A table summarizing clinical characteristics is essential.
  • Two tables with clinical characteristics were added (table 1 and 2).

  • Present data on the number of recurrences, deaths, and cause-specific deaths.
  • We added number of recurrences, deaths and cause-specific deaths.

  • In Figure 2, include the number of patients at specific follow-up intervals (e.g., 12, 24, 36 months).
  • We added the number of patients at specific follow-up intervals in figure 2.

  • Concerns regarding Table 1:
  1. Clarify the statement regarding ">60 months median survival." Given the study commenced in January 2015, and the longest follow-up was approximately 9 years (108 months), a median survival time of 160 months seems inaccurate.
  • Consider presenting the median follow-up time and outcome numbers for each subgroup instead.
  • Sorry for the mistake. We corrected the median survival time and added the median follow-up time in table 1 (now table 3).

  • For Figure 3:
  • Avoid displaying results for subtypes with fewer than five patients to prevent confusion.
  • Include patient counts at specific follow-up intervals for each group.
  • Subtypes with less than 5 subjects were excluded from figure 3. Patient count at specific follow-up intervals for each group was added.

  • Lines 137-139 suggest a misunderstanding of median OS, median DSS, and median DFS. Please review these definitions.
  • Sorry for the mistake. Median survivals were corrected.

  • For Figure 4, indicate patient follow-up numbers for each group at specific time points.
  • Patient count at specific follow-up intervals for each group was added.

  • Regarding Figure 5:
  • Correct the legend for Figure 5.
  • If Figure 5 is largely redundant with Figure 3, consider simplifying or omitting it.
  • Figure 5 was removed.

  • Lines 154-156: Given these acknowledgments, multivariable Cox regression analysis becomes necessary for meaningful comparisons.
  • Cox regression was performed and added to the paper (table 4).

  • Discussion:
  1. Summarize the key findings with reference to the study's objectives.
  • The key findings with reference to real life practice were specified at the end of the Discussion.

  • The discussion should be focused, driven by a central thesis based on the study's primary objectives and findings.
  • The discussion was focused on the impact of the results on clinical practice (the primary aim was to evaluate real life data).

  • Consider including the following reference for discussion: “Wu CN, Chuang IC, Chuang MJ, Chen WC. "Sinonasal renal cell-like adenocarcinoma: An Easily Misdiagnosed Sinonasal Tumor." This could provide valuable context and comparison to your findings.
  • This reference was added.

Reviewer 2 Report

Comments and Suggestions for Authors

This manuscript provides valuable insights into the prognosis of sinonasal tumors. However, to further refine and strengthen your manuscript, please consider the following recommendations:

·        Please detail the criteria for patient selection and exclusion more explicitly to clarify the study population

·        Consider elaborating on the statistical methods, especially the rationale behind choosing specific tests and any assumptions made that could affect the interpretation of results

·        Adding comparative data from other regions or studies could contextualize your findings within the global research landscape on sinonasal tumors

·        Addressing the potential for selection bias and its impact on study results could strengthen the study's validity

·        Enhancing the discussion on the clinical implications of your findings for practitioners in the field would be beneficial

Author Response

Dear Reviewer,

We thank you for the opportunity to revise our paper. Hereby you will find attached a point-by-point answer to each suggestion. Changes were highlighted.

  • This manuscript provides valuable insights into the prognosis of sinonasal tumors. However, to further refine and strengthen your manuscript, please consider the following recommendations:
    • Please detail the criteria for patient selection and exclusion more explicitly to clarify the study population
  • Thanks for your positive comments and suggestions. The inclusion criteria were better reported in the Materials and methods section to clearly clarify the study population.

  • Consider elaborating on the statistical methods, especially the rationale behind choosing specific tests and any assumptions made that could affect the interpretation of results
  • Multivariate analysis was added and the rationale was explained.

  • Adding comparative data from other regions or studies could contextualize your findings within the global research landscape on sinonasal tumors.
  • Geographical areas of other studies cited in the discussion were added.

  • Addressing the potential for selection bias and its impact on study results could strengthen the study's validity
  • The potential for selection bias was discussed. Since all consecutive patients with malignant sinonasal tumor were included, the risk of selection bias was minimal and this represents a strength of the study.

  • Enhancing the discussion on the clinical implications of your findings for practitioners in the field would be beneficial
  • The clinical implications of our findings for practitioners was added at the end of the Discussion.

We remain at your disposal for any further clarification.

Best regards,

The Authors

Round 2

Reviewer 1 Report

Comments and Suggestions for Authors

Major consideration:

It is inappropriate to use 5-year OS/DSS/DFS throughout the document. According to the follow-up data at each time point shown in Figure 2's Kaplan-Meier curve, only 13 individuals were followed up to 48 months. It is questionable to calculate 5-year survival rates when the follow-up duration for most participants does not reach 5 years. Similarly, those not reaching the 5-year follow-up should not be included in the numerator or denominator for calculating 5-year survival rates. The same logic applies to 2-year OS/DSS/DFS calculations.

Abstract:

Line 19-20: Please change to “The Cox multivariate regression analysis did not reveal any statistically significant effects of age, stage, or histology on survival outcomes.”

Introduction:

The content currently presented in lines 59-61 appears out of place. It may be more appropriate to relocate this section to the concluding part of the second paragraph, as it seems to address a knowledge gap.

Results:

1.     Combine Tables 1 and 2 into a single table for streamlined presentation.

2.     Remove the redundancy in lines 111-113 as the information duplicates what is presented in Table 2.

3.     Lines 132-135 fail to address the issue of loss to follow-up, which is a crucial aspect for the integrity of the study's findings. Could you please confirm this?

4.     Integrate Figure 2 and its corresponding data table (Fig 2d) to enhance readability and interpretation.

5.     Regarding Table 3, an explanation is needed for the discrepancy where the median follow-up time is listed as 8 months, yet the median survival time is 84 months for Keratinizing SCC. Please clarify this in your data.

6.     Revise Table 3 and lines 169-170: If more than 50% of patients were disease-free at the five-year follow-up, the term 'median survival time' should be replaced with 'not estimable.' Additionally, if a high rate of censoring occurred (i.e., fewer than half of the events were observed), 'median survival time' should not be used as the primary outcome.

7.     Figures 3 and 4 exhibit the same issue as previously noted with Figure 2. Please address this.

8.     Review the data presented in Table 4 regarding OS and DSS stages. The figures shown (e.g., <0.001 (<0.001->100)) appear to be incorrect. If the data is not applicable, please use "N/A". For the DSS specifically, if the event numbers were too small to yield stable model results, consider removing the DSS data from this table.

Discussion:

The discussion section currently appears overly lengthy. Once again, I recommend that it be more focused and clearly driven by a central thesis that aligns with the study’s primary objectives and findings.

Author Response

Dear Reviewer,

We thank you for the opportunity to revise our paper. Hereby you will find attached a point-by-point answer to each reviewers’ suggestion. Changes were highlighted.

  • Major consideration: It is inappropriate to use 5-year OS/DSS/DFS throughout the document. According to the follow-up data at each time point shown in Figure 2's Kaplan-Meier curve, only 13 individuals were followed up to 48 months. It is questionable to calculate 5-year survival rates when the follow-up duration for most participants does not reach 5 years. Similarly, those not reaching the 5-year follow-up should not be included in the numerator or denominator for calculating 5-year survival rates. The same logic applies to 2-year OS/DSS/DFS calculations.
  • Thanks for your comments and suggestions. We used 2-year OS/DFS/DSS throughout the paper, instead of 5-year OS/DFS/DSS (e.g., in Discussion and Conclusions). Patients not reaching 2- or 5-year follow-up were not included for calculating the rates.

  • Abstract: Line 19-20: Please change to “The Cox multivariate regression analysis did not reveal any statistically significant effects of age, stage, or histology on survival outcomes.”
  • We changed the sentence.

  • Introduction: The content currently presented in lines 59-61 appears out of place. It may be more appropriate to relocate this section to the concluding part of the second paragraph, as it seems to address a knowledge gap.
  • We moved the content.

  • Results: 1. Combine Tables 1 and 2 into a single table for streamlined presentation.
  • We combined tables 1 and 2.

  • Remove the redundancy in lines 111-113 as the information duplicates what is presented in Table 2.
  • We removed the redundancy.

  • Lines 132-135 fail to address the issue of loss to follow-up, which is a crucial aspect for the integrity of the study's findings. Could you please confirm this?
  • No patient was lost to follow-up.

  • Integrate Figure 2 and its corresponding data table (Fig 2d) to enhance readability and interpretation.
  • We integrated Figure 2.

  • Regarding Table 3, an explanation is needed for the discrepancy where the median follow-up time is listed as 8 months, yet the median survival time is 84 months for Keratinizing SCC. Please clarify this in your data.
  • The median follow-up time was shorter of median survival time because most patients were diagnosed and treated in the last part of the recruitment period (2015-2023).

  • Revise Table 3 and lines 169-170: If more than 50% of patients were disease-free at the five-year follow-up, the term 'median survival time' should be replaced with 'not estimable.' Additionally, if a high rate of censoring occurred (i.e., fewer than half of the events were observed), 'median survival time' should not be used as the primary outcome.
  • We wrote that median DFS was not estimable for early tumors. We reported median survival time but it was not the primary outcome of the analyses.

  • Figures 3 and 4 exhibit the same issue as previously noted with Figure 2. Please address this.
  • We corrected figure 3 and 4.

  • Review the data presented in Table 4 regarding OS and DSS stages. The figures shown (e.g., <0.001 (<0.001->100)) appear to be incorrect. If the data is not applicable, please use "N/A". For the DSS specifically, if the event numbers were too small to yield stable model results, consider removing the DSS data from this table.
  • We reviewed the data in table 4. DSS data were removed from the table.

  • Discussion: The discussion section currently appears overly lengthy. Once again, I recommend that it be more focused and clearly driven by a central thesis that aligns with the study’s primary objectives and findings.
  • We are sorry for the long discussion. However, the journal ask us to write at least 4000 words.

We remain at your disposal for any further clarification.

Best regards,

The Authors